

# Cell wall composition and lignin biosynthetic gene expression along a developmental gradient in an Australian sugarcane cultivar

William P. Bewg[1] and Heather D. Coleman[2]

[1] Centre for Tropical Crops and Biocommodities, Queensland University of Technology, Brisbane, Queensland, Australia
[2] Department of Biology, Syracuse University, Syracuse, NY, United States of America

## ABSTRACT

Sugarcane bagasse is an abundant source of lignocellulosic material for bioethanol production. Utilisation of bagasse for biofuel production would be environmentally and economically beneficial, but the recalcitrance of lignin continues to provide a challenge. Further understanding of lignin production in specific cultivars will provide a basis for modification of genomes for the production of phenotypes with improved processing characteristics. Here we evaluated the expression profile of lignin biosynthetic genes and the cell wall composition along a developmental gradient in KQ228 sugarcane. The expression levels of nine lignin biosynthesis genes were quantified in five stem sections of increasing maturity and in root tissue. Two distinct expression patterns were seen. The first saw highest gene expression in the youngest tissue, with expression decreasing as tissue matured. The second pattern saw little to no change in transcription levels across the developmental gradient. Cell wall compositional analysis of the stem sections showed total lignin content to be significantly higher in more mature tissue than in the youngest section assessed. There were no changes in structural carbohydrates across developmental sections. These gene expression and cell wall compositional patterns can be used, along with other work in grasses, to inform biotechnological approaches to crop improvement for lignocellulosic biofuel production.

## INTRODUCTION

Sugarcane is a C4 perennial grass of high economic importance in many parts of the world (*Suprasanna et al., 2011*). In addition to the production of high levels of sucrose in the stem, it produces large amounts of lignocellulosic biomass that has the potential to be used for the production of bioethanol (*Canilha et al., 2012*). Sugarcane is a particularly attractive source of biomass for lignocellulosic biofuels production, as it is already transported to a central location for the production of sugar. As a result, the bagasse remaining following sugar production is does not require transport costs that otherwise represent a significant cost in bioethanol production. However, as with all potential lignocellulosic feedstocks, the recalcitrance of the biomass presents challenges that need to be addressed.

Corresponding author
Heather D. Coleman,
hcoleman@syr.edu

The deposition of the secondary cell wall is an important step in terrestrial plant development (*Weng & Chapple, 2010*), involving the ordered deposition of cellulose and hemicellulose followed by the impregnation of lignin polymers into this polysaccharide matrix (*Vogel, 2008*). Lignin polymers are comprised of guaiacyl (G), syringyl (S) and *p*-hydroxyl-phenyl (H) units, through oxidative polymerization of coniferyl, sinapyl and *p*-coumaryl alcohols respectively, that are produced through the lignin biosynthesis pathway (*Boerjan, Ralph & Baucher, 2003*; *Liu, 2012*).

Due to the importance of lignin in structural stability and water transportation, the role and function of each gene within the lignin biosynthesis pathway is well established (*Boerjan, Ralph & Baucher, 2003*; *Bonawitz & Chapple, 2010*). The relationship between lignin and efficiency of lignocellulosic bioethanol production has led to increased focus into lignin biosynthesis and manipulation, and advances the possibility of cost-competitive bioethanol being produced from lignin-altered sugarcane bagasse. Given the influence lignin has on cell wall digestibility, further understanding of control and timing of lignin deposition will be applicable for the genetic modification of plants to specifically alter lignin characteristics.

While a number of studies have looked at cell wall formation in sugarcane previously (*Bottcher et al., 2013*; *De Souza et al., 2013*; *Lingle & Thomson, 2012*), here we aim to assess the expression profile of lignin biosynthetic genes and cell wall composition of a commercially relevant Australian sugarcane cultivar (KQ228). In particular, we look at a developmental gradient to further understand the relationship between gene expression and cell wall formation and composition, with the goal of providing critical information for the biotechnological development of improved varieties of sugarcane for second generation biofuel production.

## MATERIALS AND METHODS

### Gene identification

Primers were designed using sequences available from the NCBI database (Table 1). Not all genes had annotated accessions available and consensus sequences were assembled from the sugarcane EST database after BLAST analysis with the equivalent maize gene as a reference sequence. The final consensus sugarcane sequences were created using only sugarcane EST sequences. Amplicons of all lignin biosynthesis genes were run on a high-resolution gel to confirm there was only one product, and then sequenced to determine primer specificity before use in qRT-PCR.

### Plant material, growth conditions and tissue collection

Sugarcane generated from callus (cultivar KQ228, generously provided by BSES Ltd, Meringa Queensland) was acclimatized in growth chambers before being transferred to a greenhouse. KQ228 represents a commercially important smut-resistant cultivar in Queensland. Plants were grown for nine months to the full height attainable in the greenhouse before being destructively harvested for analysis. Each plant was divided into five different sections A–E to represent increasing tissue maturity, with Section A being the youngest tissue and Section E being the most mature tissue (Fig. 1). Within each section,

**Table 1** qRT-PCR primers designed for the quantification of expression levels of lignin biosynthesis pathway genes.

| Amplicon | Forward (5′–3′) | Reverse (5′–3′) | Size (bp) | Maize Accession/ EST numbers | Sugarcane Accession/ EST numbers | Related homologue in *Bottcher et al. (2013)* |
|---|---|---|---|---|---|---|
| β-Tubulin[1] | GGAGGAGTACCCTGACAGAATGA[a] | CAGTATCGGAAACCTTTGGTGAT[a] | 68 | | CA222437[b] | N/A |
| *PAL* | GACATCCTGAAGCTCATGTCG | ACCGACGTCTTGATGTTCTCC | 92 | | EF189195 | *PAL1* |
| *C4H* | GTTCACCGTGTACGGCGACCACT | GAAGAAGGGCACCGTCATGATCC | 61 | AY104175 | CA131376; CA146299; CA196076; CA137884; CA263105 | *C4H1* |
| *4CL* | CTTCCCGACATCGAGATCAACAAC | CTCATCTTCCCGAAGCAGTAGGC | 62 | AY566301; AX204868 | CA184118; CA215779; CA136560; CA176600; CA135257 | *4CL1* |
| *C3H* | GTCGACGAGCAGGTCTTCAAAGC | CGTGCTCCTCCATGATCTTCAC | 73 | AY107051; BT086560 | CA262303; CA247763 | *C3H2* |
| *CCoAOMT* | ACCTCATCGCAGACGAGAAGAAC | AGCCGCTCGTGGTAGTTGAGGTAG | 91 | AJ242980; EU952463; NM_001158013 | 5′ end: CA168805; CA071322; CA159865; CA180815 3′ end: CA159865; CF575000; CA279207; CA179873 | *CCoAOMT1* |
| *CCR* | AGCAGCCGTACAAGTTCTCG | GAAGGTTCTTCACCGTGTCG | 96 | | AJ231134 | No Match |
| *F5H* | GGTTCATCGACAAGATCATCGAC | GTCGGGGCTCTTCCCGCGCTTCAC | 53 | AX204869 | 5′ end: CA185931; CA134666; CA135938 3′ end: CA287472; CA278023; CA253395; CA103877 | *F5H1* |
| *COMT* | TACGGGATGACGGCGTTCGAGTAC | GTGATGATGACCGAGTGGTTCTT | 92 | | AY365419; AJ231133 | *COMT1* |
| *CAD* | ATCAGCTCGTCGTCCAAGAAG | ACCGTGTCGATGATGTAGTCC | 128 | | AJ231135 | *CAD2* |

**Notes.**
[a] From *Rodrigues, De Laia & Zingaretti (2009)*.
[b] All EST sequences with the prefix '*CA*' are from the Sugarcane Expressed Sequence Tag project (SUCEST).

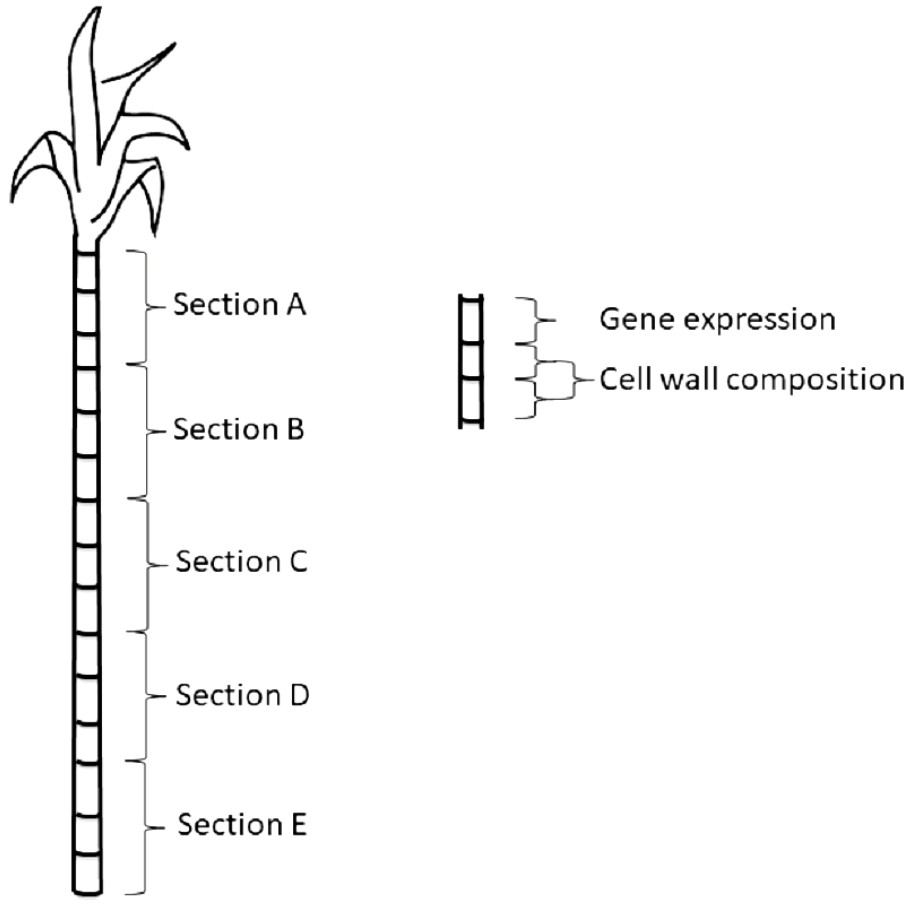

**Figure 1** **Schematic of sampling sections along the sugarcane stem.** This schematic shows the sections that are discussed in all other figures and tables with Section A being the youngest tissue and Section E being the most mature tissue.

there were three nodes, and the topmost node was used for qRT-PCR with the remaining two nodes used for cell wall composition.

Harvesting occurred between 10 am and 2 pm in a single session to minimize light or circadian related fluctuations in gene expression levels (*Rogers et al., 2005*). For all stem analyses, only internode tissue was used and node tissue was discarded. The root ball was washed to remove potting mix and buttress roots (*Moore, 1987*) were collected from each plant for qRT-PCR analysis. Roots were included in the development of this profile to begin to gain a general understanding of overall lignin biosynthetic gene expression in this tissue.

## RNA extraction and qRT-PCR

RNA was extracted from all tissue samples using Tri Reagent (Sigma, St. Louis, MO, USA). Integrity of RNA was confirmed by 2% agarose gel, and RNA concentrations quantified with a Nanodrop 2000 spectrophotometer. A total of 1 µg of RNA was treated with RQ1 RNase-free DNase (Promega, Madison, WI, USA). DNase-treated RNA was used as a template for first strand cDNA synthesis using M-MLV Reverse Transcriptase (Promega,

Madison, WI, USA). RT negative samples were prepared by replacing reverse transcriptase with water.

qRT-PCR was optimized to attain suitable $R^2$ and PCR efficiency values, and primers were validated against the housekeeping primers to ensure comparable rates of product amplification (*Livak & Schmittgen, 2001*). qRT-PCR utilized GoTaq qRT-PCR Master Mix (Promega) in a 20 µL total reaction volume with 20 ng of cDNA template using 10 mM forward and reverse primers. Samples were prepared by a CAS1200 robot (Corbett, Uithoorn, the Netherlands) and analyzed using a Rotor-Gene Q (Qiagen, Hilden, Germany) Relative transcript levels were quantified using delta critical threshold values (ΔCt) as previously described (*Levy, Edelbaum & Sela, 2004*) using β-tubulin as the housekeeping gene (*Rodrigues, De Laia & Zingaretti, 2009*).

### Cell wall composition

Tissue for cell wall compositional analysis was prepared as previously described (*Hames et al., 2008*). Dried samples were milled to pass through a 2 mm screen and extracted overnight with water and ethanol respectively (*Sluiter et al., 2008c*). A sample of this prepared material was dried overnight at 105 °C and used to determine the total solids (*Sluiter et al., 2008a*).

Cell wall composition was quantified by a modified acid hydrolysis method (*Sluiter et al., 2008b*) using 0.125 g biomass, 1.5 mL 72% sulfuric acid and 42 mL of water. This acid hydrolysis method was selected as it has been noted to provide an accurate assessment of lignin quantity (*Jung et al., 1999*). Samples were incubated at room temperature for two hours with regular mixing prior to hydrolysis at 121 °C for one hour. The hydrolysate was filtered through medium glass crucibles and the acid insoluble lignin determined gravimetrically. Acid soluble lignin remaining in the hydrolysate was determined by UV-spectrophotometry. Cell wall carbohydrates, hydrolyzed into their individual monomers, were analyzed using High Performance Liquid Chromatography. A Waters e2695 Separations Module and Showa Denko Shodex SP-0810 sugar column (85 °C) with micro-guard de-ashing columns equipped with a Waters 2414 Refractive Index Detector were employed.

## RESULTS

### qRT-PCR expression profiles of lignin biosynthesis genes

Expression profiles for the nine lignin biosynthesis genes were established after qRT-PCR analysis of the five stem sections and the root tissue (Figs. 1, 2 and 3, Fig. S1). The ΔCt values were normalized against Section A to allow for easier comparison of changes in expression in relation to young tissue for each gene. The raw ΔCt values show that in Section A, phenylalanine ammonium lyase (*PAL*) is expressed at levels greater than the other eight lignin biosynthesis genes analyzed (Table S1). Caffeoyl-CoA O-methyltransferase (*CCoAOMT*), caffeic acid O-methyltransferase (*COMT*) and cinnamoyl-CoA reductase (*CCR*) also had greater expression levels in Section A than cinnamyl alcohol dehydrogenase (*CAD*), 4-coumarate:CoA ligase (*4CL*), cinnamate 4-hydroxylase (*C4H*), ferulate 5-hydroxylase (*F5H*) and *p*-coumarate 3-hydroxylase (*C3H*). These trends are seen across the developmental gradient (Table S1).

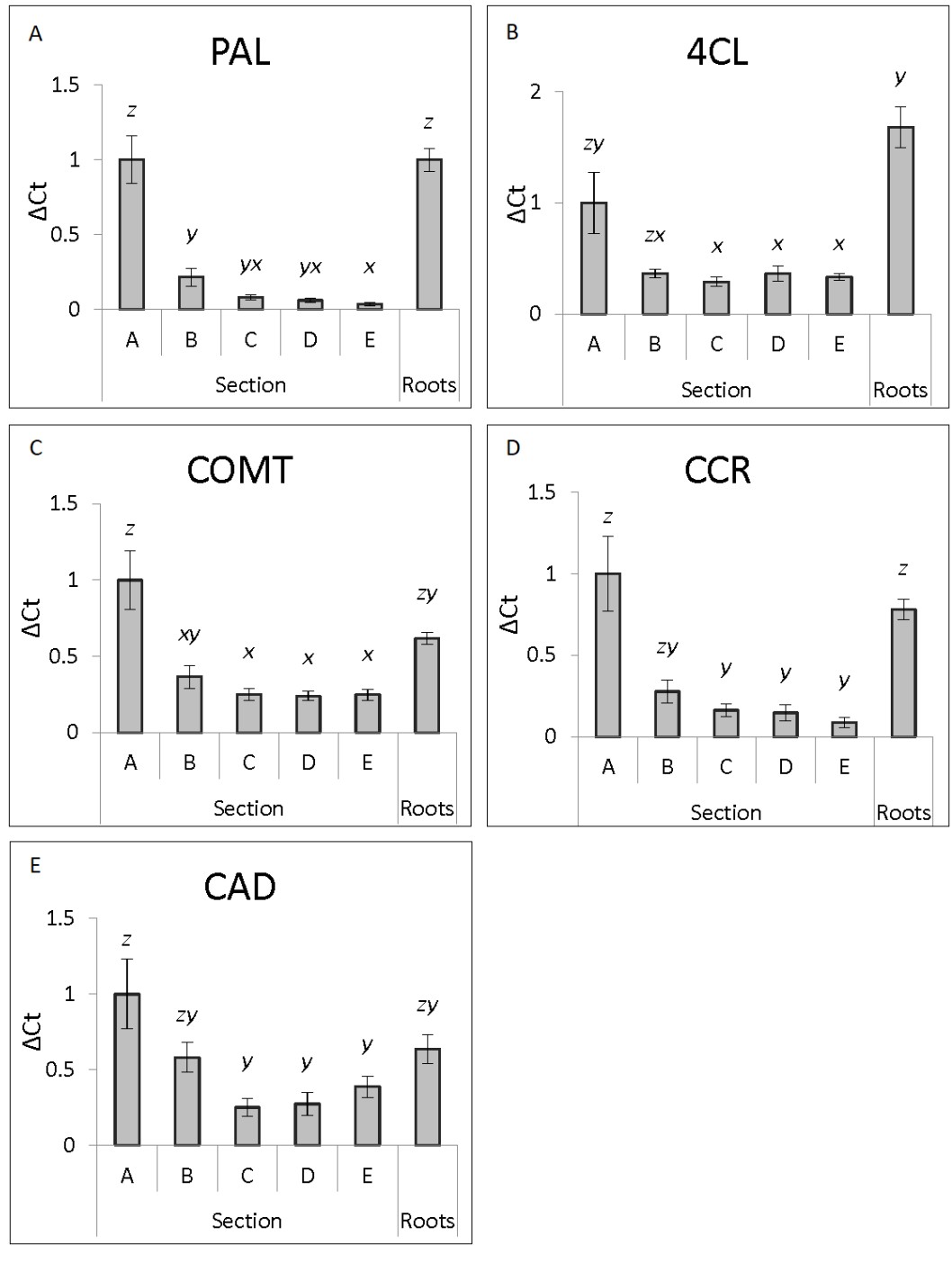

**Figure 2    Genes with decreasing expression along developmental gradient.** Genes showing highest expression in section A (young stem tissue) with decreased expression in more mature stem regions. $\Delta$Ct expression levels of lignin biosynthesis genes from the five stem sections and roots ($n = 5$ individual plants per tissue section) normalized against section A for each individual gene is shown with standard error of the mean. Statistical differences are noted by different letters above bars ($x$, $y$ and $z$) after ANOVA analysis with Tukey post-hoc analysis ($p \leq 0.05$).

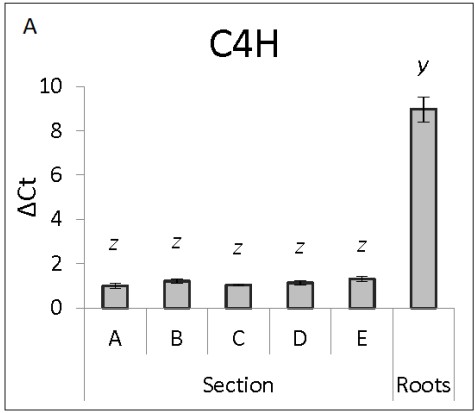
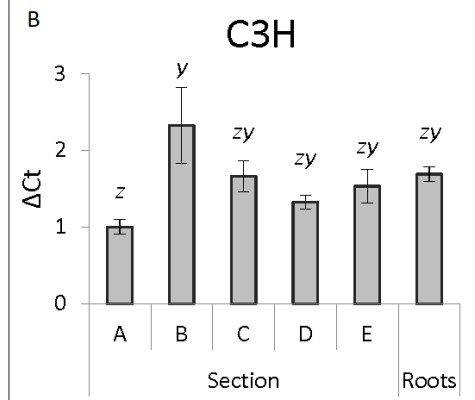
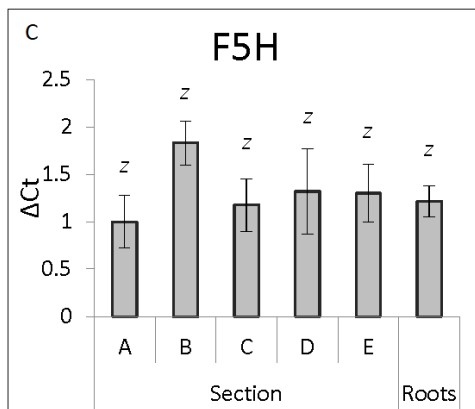
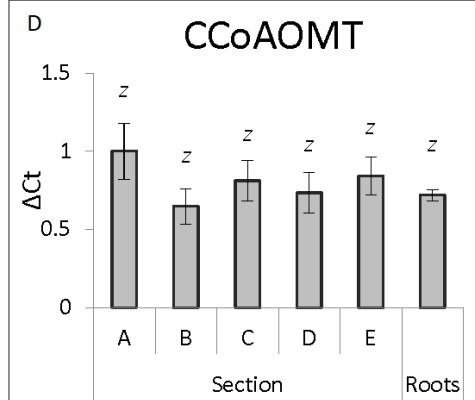

**Figure 3** **Genes showing consistent expression along developmental gradient.** $\Delta$Ct expression levels of lignin biosynthesis genes from the five stem sections and roots ($n = 5$ individual plants per tissue section) normalized against section A for each individual gene is shown with standard error of the mean. Statistical differences are noted by different letters above bars ($x$, $y$ and $z$) after ANOVA analysis with Tukey post-hoc analysis ($p \leq 0.05$).

The genes analyzed in stem tissue were separated into two general categories, those with highest expression in Section A that decreased as stem tissue matured (*PAL, CCR, 4CL, COMT* and *CAD*) (Fig. 2), and those where expression showed little change across developmental stages (*C3H, F5H, C4H* and *CCoAOMT*) (Fig. 3). The expression pattern of the genes, and their subsequent division into two groups, was not consistent with their position within the lignin biosynthetic pathway (Fig. S1).

Among the genes with highest expression in Section A (Fig. 2), the difference between Section A and the other sections is largest in *PAL* and *CCR*, with expression in Section B decreased by 70%–80% and by more than 90% in Section E. *4CL* and *COMT* show a pronounced decrease in expression, with only a 65% decrease from Section A to Section B, with minor expression decreases in older sections (Fig. 2). *CAD*, the final gene in this group, showed the lowest expression difference among all five sections analyzed, with difference variation ranging from 25 to 40% (Fig. 2). Only *PAL* and *COMT* show a significant

**Table 2 Accumulation of individual cell wall components in stem regions of increasing maturity.** The percentage of each component of the total composition is shown with the standard error of the mean. Values in bold type are significantly different (ANOVA analysis with Tukey posthoc analysis, $p \leq 0.05$) to section A for each component. No significant differences were observed between sections B–E for any component. $n = 5$.

| Section | Total lignin | | Acid insoluble lignin | | Acid soluble lignin | | Glucose | | Xylose | | Galactose | | Arabinose | |
|---|---|---|---|---|---|---|---|---|---|---|---|---|---|---|
| | % | +/− | % | +/− | % | +/− | % | +/− | % | +/− | % | +/− | % | +/− |
| A | 20.76 | 0.52 | 15.57 | 0.48 | 5.19 | 0.11 | 49.26 | 0.53 | 20.58 | 0.35 | 0.32 | 0.19 | 2.28 | 0.09 |
| B | **22.35** | **0.27** | **17.14** | **0.32** | 5.21 | 0.06 | 47.93 | 0.50 | 20.01 | 0.29 | 0.00 | 0.00 | **1.63** | **0.05** |
| C | **23.39** | **0.13** | **18.29** | **0.10** | 5.10 | 0.08 | 48.09 | 0.49 | 20.36 | 0.35 | 0.00 | 0.00 | **1.58** | **0.04** |
| D | **23.24** | **0.29** | **18.08** | **0.33** | 5.16 | 0.06 | 48.00 | 0.54 | 20.90 | 0.27 | 0.00 | 0.00 | **1.58** | **0.06** |
| E | **22.49** | **0.24** | **17.40** | **0.30** | 5.09 | 0.06 | 47.34 | 0.18 | 21.58 | 0.46 | 0.10 | 0.09 | **1.81** | **0.12** |

reduction in expression in Sections B when compared to Section A, but all five genes have significantly lower expression in Section C, D and E relative to Section A.

The second group shows similar expression across all five sections of stem tissue analyzed (Fig. 3). *C4H*, *CCoAOMT* and *F5H* all show no significant differences in expression levels across the five stem sections. *C3H* shows a significant increase in expression levels between Section A and Section B before stabilizing in Sections C, D and E.

In root tissue, *C3H*, *CCoAOMT*, *F5H* and *CAD* expression was not significantly different to any stem Section (A–E) (Figs. 1–3, Fig. S1). Expression levels of *PAL* and *4CL* were not significantly different to Section A, but were significantly higher than Sections B–E (Fig. 2). *CCR* and *COMT* showed a similar pattern being not significantly different to Sections A or B but significantly higher than expression in Sections C–E (Fig. 2). Expression of *C4H* in root tissue was approximately 9-fold higher than in any stem section (Fig. 3).

## Cell wall compositional analysis

Cell wall composition was quantified in the five stem sections (Table 2). Section A had significantly lower lignin levels than the more mature stem internodes, though levels appear to stabilize after Section B. This was due to lower acid insoluble lignin in Section A as there are no significant differences in acid soluble lignin levels across the five stem sections. Glucose, xylose and galactose did not vary across the developmental gradient, but arabinose levels were higher in Section A than in more mature tissue. There was an inverse correlation between total lignin content and arabinose content ($R^2 = 0.89$). This correlation was consistent with acid insoluble lignin, but did not hold with acid soluble lignin.

## DISCUSSION

Sugarcane bagasse has great potential as a lignocellulosic biofuels source, in part, due to its already being moved to a centralized location for sugar production. In order to effectively produce fermentable sugars from bagasse, the challenge of cell wall recalcitrance needs to be overcome. Improved understanding of lignin biosynthesis and deposition in sugarcane will be of great value when deciding the most appropriate approaches to facilitate the development of commercial lines with increased saccharification potential. The work herein uses an economically important smut-resistant Australian sugarcane

cultivar, KQ228, and assesses lignin biosynthetic gene expression and cell wall composition along a developmental gradient in an attempt to further characterize the timing and location of lignin deposition to guide attempts to improve bagasse for lignocellulosic biofuels production. It has been shown that disease resistance and lignin content are often related (*Cass et al., 2015*; *Yang et al., 2017*), making this an attractive cultivar for this work.

Whereas in our study we look only at one specific homologue for each lignin biosynthetic gene, we have provided a comparison to previous work (*Bottcher et al., 2013*) wherein multiple homologues are assessed (Table 1). Our work was based on available genome data at the time of the study, and despite being somewhat limited relative to *Bottcher et al. (2013)*, provides confirmation and comparison with another economically important cultivar. In addition to this, we also examined lignin gene expression in buttress roots as this can be a key storage sink for carbon.

The trends in the stem expression data dichotomize the lignin biosynthesis genes. The two 'expression pattern groups' are genes for which expression decreases with tissue age (*PAL*, *CCR*, *4CL*, *COMT* and *CAD*) or genes for which expression remains constant during maturation (*C3H*, *F5H*, *C4H* and *CCoAOMT*). As *PAL* catalyzes the entry of metabolites into the lignin biosynthesis pathway (*Liu, 2012*; *Weng & Chapple, 2010*), its high level of expression in younger tissue found in this study may represent an initial metabolic flux to provide a burst of metabolites for the various phenylpropanoid pathways including lignin biosynthesis. *CCR* functions in the final stages of lignin biosynthesis and is considered a committed step, key in the production of the individual lignin monomers (*Vogt, 2010*; *Weng & Chapple, 2010*). Given the position of *CCR* in the lignin biosynthesis pathway, it may act as a regulating control point for directing the metabolic flux into lignin monomer production (*Lacombe et al., 1997*). As high expression of *PAL* in young tissue may act to stimulate metabolic flux into phenylpropanoid production, high expression of *CCR* in young tissue may ensure a high level of metabolite commitment into lignin biosynthesis, which is fundamentally important for healthy plant development (*Weng & Chapple, 2010*).

The expression profiles of *4CL* and *COMT* are similar to that of *PAL* and *CCR*, but they retain slightly higher expression levels in more mature tissue. *4CL* represents an important branch where metabolites are directed either into lignin biosynthesis or to alternative phenylpropanoid biosynthesis pathways (*Vogt, 2010*; *Weng & Chapple, 2010*). Its position also allows for direct metabolite contribution into H monomer biosynthesis or redirection of metabolites for G or S monomer biosynthesis. The high level of *4CL* expression in young tissue may reflect its response to the metabolic flux into the phenylpropanoid pathway initiated by *PAL*. *COMT* is the last of two enzymes entirely responsible for the production of the S lignin monomer within the lignin biosynthesis pathway (*Bonawitz & Chapple, 2010*). The increased expression of *COMT* in young tissue in this research may be to ensure S monomer production during the availability of the initial metabolic flux. Previous work has shown that the RNAi suppression of *COMT* in sugarcane resulted in decreased lignin and altered S:G ratio (*Bewg et al., 2016*).

The final gene showing a reduction in expression as stem tissue matures was *CAD*, though the trend was not as strong as the previously discussed genes. *CAD* represents the final enzyme in the lignin biosynthesis pathway catalyzing the production of precursor

monolignols and committing them to lignin monomer synthesis (*Ferrer et al., 2008*). The initial high expression of *CAD* in young tissue may relate to the increased metabolic flux through the lignin biosynthesis pathway. Whereas overall trends between our work and previous research are the same for these genes with decreasing expression for increasing maturity, there are differences, particularly with *CAD* and *COMT*. The discrepancies between the current and published research may be a result of various experimental differences between the current research and published findings, but it is more likely that the differences arise from the differences in cultivars.

Three genes were identified with relatively consistent expression across the maturity gradient: *C4H*, *F5H*, and *CCoAOMT*. Results for *C4H* were consistent with other results (*Papini-Terzi et al., 2009*). For *F5H*, expression in the Brazilian low and high lignin cultivars was highest in intermediate aged internodes with the exception of the high lignin pith samples wherein it was highest in the mature tissue (*Bottcher et al., 2013*). In the 30 cultivars with varying Brix levels, *F5H* expression levels were higher in maturing stem tissue than in young tissue (*Papini-Terzi et al., 2009*). Our results for *CCoAOMT* closely mirrored the results of *Bottcher et al. (2013)*. We have previously published work describing the downregulation of both the *F5H* and *CCoAOMT* in sugarcane, with the result being increased glucose release by enzymatic hydrolysis but with no decrease in lignin. In the *F5H* lines this was attributed to a change in the lignin monomer ratio (*Bewg et al., 2016*)

The final gene assessed in our study, *C3H*, had the highest expression level in Section B, immediately below the most juvenile Section A. *C3H* catalyzes the second aromatic hydroxylation reaction in the lignin biosynthesis pathway and is an important hub in controlling metabolic flux into G and S lignin monomer synthesis (*Barriere et al., 2004*; *Weng & Chapple, 2010*). *CCoAOMT*, along with *C3H*, is hypothesized to be important control points for cell wall lignification by acting as part of the ferulate production pathway (*Barriere et al., 2004*). *CCoAOMT* is responsible for the 3′ methylation of caffeoyl-CoA to produce feruloyl-CoA, a key step in the production of G and S lignin monomers (*Hisano, Nandakumar & Wang, 2009*; *Raes et al., 2003*). The feruloyl residues aid in cross-linking within the cell wall and may increase the resistance of the cell wall to hydrolysis by adding to its structural stability (*Barriere et al., 2004*; *Bonawitz & Chapple, 2010*; *Grabber, 2005*). The relatively steady expression of *CCoAOMT* and *C3H* within the maturing sugarcane stem may reflect their continued requirement for feruloyl residue production for ongoing cell wall lignification and not just their role in lignin monomer biosynthesis.

Hydroxycinnamoyl transferase (*HCT*) was not included in qRT-PCR analysis as at the time of this study a specific sequence could not be confidently identified. At that time, only one published accession for sugarcane *HCT* was found (CA210265) (*Casu et al., 2007*). When analyzed by BLAST it showed very close alignment with *Zea mays* anthranilate N-benzoyltransferase (NM_001153992) (*Soderlund et al., 2009*). Further BLAST searching in the sugarcane nucleotide and EST databases of NCBI with alternative *HCT* sequences from maize (AY109546, DR807341) (*Barrière et al., 2007*) and from MAIZEWALL (2478084.2.1_REV, 2619423.2.1) (*Guillaumie et al., 2007*), *Medicago sativa* L. (AJ507825) (*Shadle et al., 2007*), *Nicotiana benthamiana* (AJ555865) (*Hoffmann et al., 2004*), *Coffea arabica* (AM116757) (*Salmona et al., 2008*) and *Triticum aestivum* L.

(CK193498, CK199765) (*Bi et al., 2011*) did not highlight any potential sugarcane *HCT* sequences, nor any conserved regions of sufficient length to design primers (standard or degenerate) for potential use in sugarcane.

To our knowledge, this is the first paper that has looked at lignin biosynthetic gene expression in sugarcane buttress roots. There were no significant differences in expression levels between root tissue and the five stem sections (A–E) for *C3H*, *CCoAOMT*, *F5H* and *CAD*. Of these, *C3H*, *CCoAOMT* and *F5H* are all in the group with plateaued gene expression during development and may highlight the promoters of these three genes as potential biotechnological tools to drive continuous and even expression of transgenes in sugarcane stem and root tissue. The only gene with an unexpected level of expression was *C4H* that had approximately 9-fold higher expression in roots than in any stem section. This suggests that the *C4H* promoter may be useful for preferential expression of transgenes in sugarcane root tissue, however further analysis, including the functionality of this promoter in additional tissue types, such as leaves, would need to be assessed.

In addition to the assessment of lignin biosynthetic gene expression, we also examined the cell wall composition along the same developmental gradient. It is well known that the composition of the cell wall material changes as a plant matures due to secondary cell wall deposition. Following cell elongation, the secondary cell wall is formed through the deposition of cellulose and hemicellulose, followed by lignification (*Vogel, 2008*; *Weng & Chapple, 2010*). Within sugarcane, rapid elongation of young internode cells precedes cell wall thickening, including lignification (*Casu et al., 2007*). No significant differences were seen in levels of structural carbohydrates including glucose, xylose or galactose indicating that the deposition of structural polysaccharides into the secondary cell wall had also occurred before harvesting of samples (in more juvenile tissue). This is in contrast to published findings in sugarcane (*Lingle & Thomson, 2012*) and maize (*Jung & Casler, 2006*). In sugarcane, cellulose peaked and then declined below internode 5, whereas hemicellulose was highest in young tissue before reducing to a steady state (*Lingle & Thomson, 2012*). In maize, glucose content increased as tissue matured before plateauing, and hemicellulose (xylose and arabinose) decreased as tissue matured before also reaching a steady state (*Jung & Casler, 2006*). The decrease in xylose and arabinose coincided with an increase in ferulates, and the authors suggest ferulates may be replacing the xylose and arabinose within the cell wall, hence their decrease during tissue maturation (*Jung & Casler, 2006*).

Results suggest the lignin deposition was complete by Section B as lignin content plateaued and no differences were detected between Sections B through E. Other studies have also found that overall lignin content increased with tissue maturity in wheat and maize (*Jung & Casler, 2006*; *Ma, 2007*). In maize stem, lignin content decreased initially before increasing to a plateau (*Jung & Casler, 2006*). In sugarcane, marked internodes harvested over a period of twelve weeks had increased lignin content over time (*Lingle & Thomson, 2012*). In a second experiment, odd numbered internodes harvested at a single time point, showed lignin content increased with maturity, with the exception of a significant decrease in internode 3 (*Lingle & Thomson, 2012*). The results of the second experiment are similar to the maize results of *Jung & Casler (2006)*, who suggest that young maize tissue is comprised of a higher percentage of lignified protoxylem vessels than more

mature tissue, that initially results in a high lignin content in very young tissue (*Jung & Casler, 2006*). It is likely that the Section A tissue (from internodes 2 and 3) was in what is the second zone identified in the studies by *Jung & Casler (2006)* and *Lingle & Thomson (2012)*. This is supported by the results of *Bottcher et al. (2013)* who showed lower lignin levels in internode 2–4 of two sugarcane cultivars before reaching a relatively steady state lignin level for internodes 5 to 18. Only one paper has examined the lignin content of root tissue, and the authors found only small changes in lignin content in the first 5 cm of root development, with lignin levels ranging from 5–9% (*Leite et al., 2017*).

Of note was the inverse relationship seen between lignin and arabinose content. This is likely the result of the pattern of cell wall deposition, in which there is a natural progression from highly substitute arabinoxylans to less branched xylan as cells fully expand. This natural progression would also correspond with increasing lignin deposition (*Carpita, 1996*; *De O Buanafina, 2009*). This is consistent with previous results comparing cell wall properties across three Miscanthus genotypes (*De Souza et al., 2015*).

The work presented herein provides a profile of lignin biosynthetic gene expression and cell wall composition for an economically important Australian sugarcane cultivar. The results support findings of previous groups and add additional information on gene expression in sugarcane buttress roots. As a key potential biofuels crop, detailed information from multiple cultivars will help to improve the understanding of lignin and cell wall formation in this species and to inform biotechnological approaches to crop improvement.

### Funding
This work was supported by the Australian Research Council Discovery Program (Heather D. Coleman; DP1093236) and the Sugar Research and Development Corporation (Sugar Research Australia; STU-068) Scholarship Program (William P Bewg). The funders had no role in study design, data collection and analysis, decision to publish, or preparation of the manuscript.

### Grant Disclosures
The following grant information was disclosed by the authors:
Australian Research Council Discovery Program: DP1093236.
Sugar Research and Development Corporation: STU-068.

### Competing Interests
The authors declare there are no competing interests.

### Author Contributions
- William P. Bewg performed the experiments, analyzed the data, wrote the paper, prepared figures and/or tables, reviewed drafts of the paper, funding.
- Heather D. Coleman conceived and designed the experiments, wrote the paper, prepared figures and/or tables, reviewed drafts of the paper, funding.

## Data Availability

The raw data is included in the Supplemental Files.

## Supplemental Information

Supplemental information for this article can be found online at http://dx.doi.org/10.7717/peerj.4141#supplemental-information.

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
