# Peer review of "Cell wall composition and lignin biosynthetic gene expression along a developmental gradient in an Australian sugarcane cultivar"

_PeerJ, doi:10.7717/peerj.4141_

## Round 0.1 · original submission · Major Revisions

I concur with the reviewers assessment. Please address all their comments before resubmission.

Reviewer 1 ·

Basic reporting

Authors should provide the comparative analysis of lignin biosynthesis pathway genes expression data and lignin content from different cultivars available till date. This will give support to this study that how it adds additional information with the preexisting data.

Experimental design

In Materials and methods section, remove the para describing about HCT gene from line 71-82 and place it in discussion section.

Validity of the findings

Expression analyses of the genes provided by the authors did not match with the lignin content across developmental gradient. Early step genes e.g., PAL, C4H, 4CL, and C3H are also involved in biosynthesis of variety of flavonoids so their expression can not be solely related with lignin production. Downstream pathway genes e.g., CCR and CAD are highly expressed in young tissue (A), which has lesser lignin content as compared to other tissues. Similarly, CCR and CAD showed higher expression in roots but no data for cell wall component is provided. F5H and CCoAOMT are not showing any specific expression pattern across developmental tissue stages. I suggest authors to add functional validation experiments of one or more genes e.g., CCR/CAD/COMT using gene silencing or gene editing approach to support data provided in this manuscript.

Additional comments

Manuscript is written well.

·

Basic reporting

Figure 1 Should be improved as mentioned in general comments.

Experimental design

Authors should explain better why they did chose the cultivar and also why 9 months. All these information should be added to the text.

Validity of the findings

Work here analyzed did not represent any unprecedented data as commented by authors in the manuscript. By the other hand, they add some knowledge about lignin accumulation mainly in root which for me is the main result of this manuscript. Authors also elucidate the possibility to use some promoter in further genetic manipulation basing on their expression profile. Still, would be desirable if authors could add S/G ratio values in order to check if the expression of some key enzymes (such as F5H and COMT) could be related to S/G in different parts of sugarcane culm.

Additional comments

Line 38-40: Authors should explain this sentence better
Line 86: Authors should explain why they choose this stage (9 months)
Line 98-99: Authors did not mention that RNA integrity was checked before proceeding to DNAse treatment
Line 126: Authors must include in figure 1 a picture of sugarcane cultivar that they analyzed at 9 months showing details of five sections here analyzed and roots as well. This definitely will help readers to understand which part of sugar cane were studied here
Line 132: sentence should be replaced by separated into two categories
Line 135: word fall should be replaced all long the manuscript. Inadequate word
Line 137: Change word ‘Of’ for ‘Among’
Line 139-143: Rewrite the sentence. CAD showed the lower expression difference among all five sections analyzed with different variation ranging from 25 to 40%. This example should also be used to rewrite previous sentences comparing results.
Line 171: This explanation should be added to M&M section to explain why use this cultivar. Authors should also explain if they expect that this smut-resistant feature could have or not any relation with lignin biosynthesis/accumulation.

Reviewer 3 ·

Basic reporting

The manuscript by Bewg and Coleman describes the patterns of gene expression related to lignin metabolism and the composition of stem and root materials of the Australian cultivar KQ228. The manuscript is well written in general and brings results that deserve to be published in my opinion.
The discussion about gene expression is well balanced in itself. However, the authors failed in contextualizing the existent knowledge about cell walls. As a result, the manuscripts end up quite lignin centric. This won't be a problem if authors haven´t measured cell wall composition. In this part, the manuscript is quite weak, describing methodology poorly and not discussing it adequately in the face of the available literature.
Because authors did not discuss the carbohydrate portion of the wall at the same level they discuss lignin gene expression; the manuscript fails to take most of what the results can give.
My opinion is that the manuscript is publishable, but it needs major changes so that it reports the scientific context of sugarcane cell walls, describe methods accurately and discuss more profoundly the results in relation the available literature.

Experimental design

Preparation of material: A picture or drawing of the plant and the positions sections were made could be useful for the reader and other authors that intend to repeat the experiment.

Cell wall composition:
1) Please describe how acid hydrolysis was made instead of just citing another paper. Instead of giving the components of a solution only, please report time and temperature. Why was sulfuric acid chosen? It usually hydrolyzes cellulose and decreases the proportion of pectin and hemicellulose monosaccharides. This happened as can be seen in Table 1. What authors want to see by accessing mostly cellulose? As lignin is bound to hemicellulose – the arabinoxyl residues of arabinoxylans – wouldn´t it be better to look at the levels of arabinose and therefore use TFA instead of sulfuric acid for hydrolysis?
2) How was acid linin extracted? What acid lignin means. There are several different methods for lignin extraction. Explain why this method was chosen. The lack of adequate methodological description prevents interpretation of the results reported in Table 1. The reader would have to find literature that describes methodologies and guess what authors have used in this paper.
3) Usually, cell wall monosaccharides are analyzed on Dionex because other types of chromatography do not separate well some of them. Was the Waters chromatography efficient to separate them? It would be important to add a chromatogram as supplemental material to inform the reader about the accuracy of monosaccharide measurements.

Validity of the findings

Results

1) Didn´t authors notice that there is a strong negative correlation (r2=0.89) between total lignin and arabinose? How authors interpret that?
2) The percentage of total lignin is quite high. Could this be an artifact of the method used? Authors should comment that.
3) Authors cannot call the composition “secondary cell wall”. Hydrolysis with sulphuric acid attacks all cell walls, including the primary ones.


Discussion
For discussion on sugarcane cell wall composition, authors failed to include De Souza et al., 2013, which would be a better comparison than maize, which is relatively distant taxonomically from sugarcane and miscanthus.
Lignin (yield only) in sugarcane roots has been reported by Leite et al. 2017, Annals of Botany. The authors found rather low yields in roots of a Brazilian variety
Authors missed an important trend that is the inverse correlation between lignin and arabinose. It is accepted that ferulic acid, the holder of lignin in the wall of grasses is esterified to arabinose residues of arabinoxylans (the main hemicellulose in cell walls). (see De Souza et al., 2013, Bioenergy Research for sugarcane and De Souza et al., 2015 Journal of Exp. Botany). This could be interesting for the discussion of the manuscript.

Additional comments

The manuscript by Bewg and Coleman describes the patterns of gene expression related to lignin metabolism and the composition of stem and root materials of the Australian cultivar KQ228. The manuscript is well written in general and brings results that deserve to be published in my opinion.
The discussion about gene expression is well balanced in itself. However, the authors failed in contextualizing the existent knowledge about cell walls. As a result, the manuscripts end up quite lignin centric. This won't be a problem if authors haven´t measured cell wall composition. In this part, the manuscript is quite weak, describing methodology poorly and not discussing it adequately in the face of the available literature.
Because authors did not discuss the carbohydrate portion of the wall at the same level they discuss lignin gene expression; the manuscript fails to take most of what the results can give.
My opinion is that the manuscript is publishable, but it needs major changes so that it reports the scientific context of sugarcane cell walls, describe methods accurately and discuss more profoundly the results in relation the available literature.


Introduction


The introduction does not have a single word about the composition and structure of sugarcane cell walls (e.g., De Souza et al., 2013 Bioenergy Research). It doesn´t mention either important work performed on sugarcane lignin (e.g., Bottcher et al. 2013 Plant Physiology; Vicentini et al. 2015 PloS one). Another example regarding lignin is the work of Jung et al. 2013, Plant Biotechnology Journal.
Authors should contextualize better sugarcane cell walls in the introduction.

Methods

Preparation of material: A picture or drawing of the plant and the positions sections were made could be useful for the reader and other authors that intend to repeat the experiment.

Cell wall composition:
1) Please describe how acid hydrolysis was made instead of just citing another paper. Instead of giving the components of a solution only, please report time and temperature. Why was sulfuric acid chosen? It usually hydrolyzes cellulose and decreases the proportion of pectin and hemicellulose monosaccharides. This happened as can be seen in Table 1. What authors want to see by accessing mostly cellulose? As lignin is bound to hemicellulose – the arabinoxyl residues of arabinoxylans – wouldn´t it be better to look at the levels of arabinose and therefore use TFA instead of sulfuric acid for hydrolysis?
2) How was acid linin extracted? What acid lignin means. There are several different methods for lignin extraction. Explain why this method was chosen. The lack of adequate methodological description prevents interpretation of the results reported in Table 1. The reader would have to find literature that describes methodologies and guess what authors have used in this paper.
3) Usually, cell wall monosaccharides are analyzed on Dionex because other types of chromatography do not separate well some of them. Was the Waters chromatography efficient to separate them? It would be important to add a chromatogram as supplemental material to inform the reader about the accuracy of monosaccharide measurements.


Results

1) Didn´t authors notice that there is a strong negative correlation (r2=0.89) between total lignin and arabinose? How authors interpret that?
2) The percentage of total lignin is quite high. Could this be an artifact of the method used? Authors should comment that.
3) Authors cannot call the composition “secondary cell wall”. Hydrolysis with sulphuric acid attacks all cell walls, including the primary ones.


Discussion
For discussion on sugarcane cell wall composition, authors failed to include De Souza et al., 2013, which would be a better comparison than maize, which is relatively distant taxonomically from sugarcane and miscanthus.
Lignin (yield only) in sugarcane roots has been reported by Leite et al. 2017, Annals of Botany. The authors found rather low yields in roots of a Brazilian variety
Authors missed an important trend that is the inverse correlation between lignin and arabinose. It is accepted that ferulic acid, the holder of lignin in the wall of grasses is esterified to arabinose residues of arabinoxylans (the main hemicellulose in cell walls). (see De Souza et al., 2013, Bioenergy Research for sugarcane and De Souza et al., 2015 Journal of Exp. Botany). This could be interesting for the discussion of the manuscript.

---

## Round 0.2 · Minor Revisions

Please see if you can add Reviewer #1's comments on cell wall composition from root. Also integrate the Supplementary file 1 (table with QRTPCR primers as a regular table in the main manuscript. This is for better citation, visibility and helping automated data mining processes to find relevant data that is not hidden in the supplementary files.

Reviewer 1 ·

Basic reporting

No comments

Experimental design

No comments

Validity of the findings

No comments

Additional comments

In this revised manuscript, authors answered all the questions/comments raised by reviewers so in my opinion this manuscripts has sufficient data to publish. Authors improved this manuscript well, however; my only concern is about the English language/grammar which need to be improved for publication and it would be great if authors could provide the lignin content and cell wall composition data for roots.

·

Basic reporting

No comment

Experimental design

No comments

Validity of the findings

No comment

Additional comments

The authors have improved the previous manuscript version and also asked all the question resulting in a better version. All the doubts regarding plant material used, developmental stages used for experiments were answered.

Reviewer 3 ·

Basic reporting

Article is relevant for the area of bioenergy

Experimental design

Authors corrected the problems pointed out accordingly

Validity of the findings

Results are sound and valid. Deserve to be published

Additional comments

Authors have changed the text and properly responded the queries. In my opinion the manuscript is now acceptable for publication.

---

## Round 0.3 · accepted · Accept

Please make sure all the gene symbols and names are italicized.